# Presurgical Language Mapping: What Are We Testing?

**DOI:** 10.3390/jpm13030376

**Published:** 2023-02-21

**Authors:** Roelien Bastiaanse, Ann-Katrin Ohlerth

**Affiliations:** 1University of Groningen, P.O. Box 716, 9700 AS Groningen, The Netherlands; 2Neurobiology of Language Department, Max Planck Institute for Psycholinguistics, P.O. Box 310, 6500 AH Nijmegen, The Netherlands

**Keywords:** presurgical language mapping, nouns and verbs, brain tumor, word production, *navigated Transcranial Magnetic Stimulation (nTMS)*

## Abstract

Gliomas are brain tumors infiltrating healthy cortical and subcortical areas that may host cognitive functions, such as language. If these areas are damaged during surgery, the patient might develop word retrieval or articulation problems. For this reason, many glioma patients are operated on awake, while their language functions are tested. For this practice, quite simple tests are used, for example, picture naming. This paper describes the process and timeline of picture naming (noun retrieval) and shows the timeline and localization of the distinguished stages. This is relevant information for presurgical language testing with *navigated Magnetic Stimulation (nTMS)*. This novel technique allows us to identify cortical involved in the language production process and, thus, guides the neurosurgeon in how to approach and remove the tumor. We argue that not only nouns, but also verbs should be tested, since sentences are built around verbs, and sentences are what we use in daily life. This approach’s relevance is illustrated by two case studies of glioma patients.

## 1. Introduction

Gliomas are malignant brain tumors infiltrating healthy and functional cortical and subcortical tissue. When low grade, they grow very slowly, allowing motor and cognitive functions to migrate to other brain areas. During tumor resection, the neurosurgeon aims to save as much of this functional tissue as possible, but cannot rely on classical localization maps, because of this functional plasticity. Therefore, glioma patients with tumors in the so-called ‘eloquent areas’ are preferably operated while the patient is awake. During surgery, the neurosurgeon stimulates small areas of the cortex or subcortex with *Direct Electrical Stimulation (DES)* while the patient performs a task, for example, naming pictures. If, during stimulation, a patient cannot name a picture correctly, it is assumed that the area is involved in speech and language production and should be spared.

Awake surgery is an intense event, both for the patient and for the neurosurgical team. Therefore, neuroscientists are looking for alternative ways to localize critical cognitive functions in the affected brain. One of these ways is *navigated Transcranial Magnetic Stimulation (nTMS)*. Instead of applying an electrical current directly on the cortical and subcortical areas, an electro-magnetic current is directed through the skull onto the cortex to inhibit the area right beneath the skull, which is a harmless procedure. The principle of interpreting DES and nTMS is the same: when a cortical area is stimulated, either electrically or electro-magnetically, and a picture cannot be named, that area is involved in the language process (note that (n)TMS can also enhance functionality of cortical tissue; in that case, another current is needed; the voltage used for the studies mentioned in this paper comprises an inhibiting current).

Of course, it is important that language skills are spared when a tumor is removed: language is the basis of communication, perhaps the most important cognitive function in human beings. While counting backwards and naming the months of the year are considered old-fashioned tests nowadays, naming pictures of objects and animals is quite a common task. However, this is not quite representative for language in daily life: we do not speak only in nouns, but rather in sentences, and sentences are constructed around verbs and contain many other word classes, such as adjectives and adverbs. Now that we have the opportunity to test glioma patients non-invasively and presurgically with nTMS, a wider range of tasks may be used; although we should realize that tumor patients cannot be tested for protracted periods because of fatigue and lack of concentration. Hence, we developed a test for the production of verbs and a noun phrases (including the article) in sentence context, in which the verbs have to be inflected for person, number and time (3rd person singular present; *he writes*). Inflection of English articles and nouns is limited (only the article *a* has to be changed to *an* when the noun starts with a vowel) but in German, for the language used for the experiments described below, the article is inflected for number, gender and case. This test is called the *Verb And Noun test for Peri-Operative testing (VAN-POP* [1]; see Figure 1). The VAN-POP can be used with both DES and nTMS mapping procedures, hence during pre- and intraoperative language testing.

In order to interpret the results of such a test, it is important to understand the process of picture naming. For reasons of simplicity, we will sketch this process and show the localization and timing of object naming with single nouns guided by Levelt′s [2] model and Indefrey′s [3] meta-analysis of neuro-imaging studies on object naming, the task most often used for presurgical language testing.

## 2. The Process of Picture Naming

A test frequently used presurgically, but also in the operation room is ‘picture naming’. A picture of, for example, a cat is shown to the patient. In response, the patient should say ‘cat’. If, during stimulation of a certain area, the patient names the picture correctly, the stimulated area is supposed to be not involved in the whole process from recognizing the picture to speaking the word *cat* and is, therefore, safe to resect.

In order to name a picture, a few steps need to be carried out, and each step takes place in a different part of the brain.

### 2.1. Picture Recognition and Conceptualization

The first step is recognition of a black-and-white drawing of a cat. The next requires linking that image to the concept of a cat stored in the brain. The concept consists of information from different parts of the brain and includes the following:Visual information such as general size, color and shape of a cat, stored in the visual cortex in the occipital lobes bilaterally.Auditory information about how a cat sounds, stored in the auditory cortex in the temporal lobes, bilaterally.Olfactory information about how a cat smells, stored in the olfactory cortex in the temporal lobes, bilaterally.Tactile information about how a cat’s fur feels, stored in the sensory cortex in the parietal lobes, bilaterally.Information about a cat’s predatory nature, diet, etc.; this is stored semantic memory in the hippocampus and, probably, in the frontal and temporal lobes bilaterally.Information about personal feelings related to the concept of a cat (cat lovers vs. cat haters), stored in the limbic system, bilaterally.

This bundle of information forms one’s concept of a cat and is stored bilaterally. This implies that for matching a picture with a concept, a large part of the cortex in both hemispheres is activated. This process does not yet involve language, it is a ‘preverbal stage’.

### 2.2. Lemma Activation

Next, the lemma CAT (lemmas are presented in upper case letters) is activated, which is the first step involving language. A lemma is an abstract word form that contains information about the meaning, the word class (verb, noun, adjective) and, in case of a verb, how many entities are involved. In the verb *to swim,* for example, only an actor is involved, the swimmer, whereas in *to write,* there are two: the writer and what is written. A lemma is not a word. The lemmas for *to swim* and *to read* are similar in English and French, but the words themselves are different (and are only retrieved during the next stage).

Lemmas are stored on the basis of their meaning; that is, when a lemma is activated, lemmas close in meaning are co-activated. Then the co-activated lemma is inhibited and the target lemma ‘wins’. This can be illustrated by the fact that the first word that comes to mind when one hears ‘cat’ is ‘dog,’ at least for the majority of the people in the Western world, independent of language. When the process of activation, co-activation and inhibition is disrupted, for example in aphasia, a semantic paraphasia may be produced: a picture of a table is named as *chair*.

According to Indefrey [3], who carried out a meta-analysis of studies on word production, lemmas are stored in the middle part of the left middle temporal gyrus (see Figure 2).

### 2.3. Lexeme Retrieval

Once the lemma CAT has been retrieved, it activates the lexeme/cat/(lexemes are given between /…/). Lexemes are the underlying word forms and they are stored on the basis of their sound structure. A lemma activates the target lexeme/cat/and lexemes that are related in sound structure are co-activated:/rat/,/fat/,/cap/. Notice that the co-activated lexemes are always words, because only words are stored as lexemes. Non-words, such as ‘dat’ or ‘rof’ cannot be co-activated, because they are not in the mental lexicon that contains the lexemes.

The difference between lemmas and lexemes can also be illustrated by tests often used during awake surgery: word fluency. Although it is unclear what these tests actually measure or what is wrong when patients score lower than non-brain-damaged speakers, we do know that ‘semantic word fluency’ tests (‘name as many animals as you can in one minute’) tap into lemma retrieval, whereas ‘phonological fluency′ tests (‘name as many words beginning with *b*′) tap into lexeme retrieval.

According to Indefrey′s [3] meta-analysis, lexemes are stored in the posterior part of the left superior and middle temporal gyrus (see Figure 2).

### 2.4. Phonological Encoding

Lexemes activate the process of phonological encoding. When the lexeme has been retrieved, the correct phonemes (phonemes are those speech sounds that can distinguish one word from another, for example, /p/ and /b/) must be inserted in the correct order. Phonetic forms are given between square brackets […]. The lexeme/cat/is phonologically correctly encoded as [kæt] and not as [fæt] or [tæk]). In addition, phonological rules are applied during phonological encoding. An example of a phonological rule is that an underlying plural/s/is pronounced like [z] when the noun ends in a ‘soft’ sound, such as a vowel or/d, z, b, v, g/:/bed/ + /s/ → [bedz].

Phonological encoding is not mentioned in the meta-analysis of Indefrey [3]. However, even though the process can be differentiated (see, e.g., Den Hollander et al. [4]), its localization is controversial. The arcuate fasciculus, the superior longitudinal fasciculus and the supramarginal gyrus, all at the left side, have been hypothesized to enable this process (see Figure 2).

The output of phonological encoding is a string of phonemes (speech sounds; see Figure 2). Now articulation can be planned and programmed at the next stage.

### 2.5. Phonetic Encoding

This stage is at the interface of language and speech: the individual phonemes are retrieved and an articulation program is made for a smooth transition from one phoneme to the next and from one syllable to the next. ‘Assimilation’ of phonemes also takes place at this level. For example: the [a] of ‘cab’ is pronounced differently from the [a] of ‘cat’ due to the following/b/that requires a lengthening of the vowel.

Planning and programming of articulation takes place in Broca′s area, the left inferior frontal gyrus (see Figure 2). Still, the word is not pronounced. For this, it needs to be articulated.

### 2.6. Articulation

The final step of the naming process is articulation. The program planned at the former stage needs to be executed for the word to be pronounced. For this, the motor cortices of both hemispheres are used. Figure 2 shows the whole process.

## 3. The Time Course of Picture Naming 

Indefrey [3] not only describes the localization of picture naming, but also its time course. As can be seen in Figure 3, each step takes roughly 100 ms. In his figure, Indefrey [3] does not fill in the time course for concept recognition, but according to the text, this takes place between 100 and 150 ms. Despite the invaluable contribution of Indefrey′s [3] paper, there are one or two points of uncertainty. The first one is the fact that it is a meta-analysis. Meta-analysis allows for comparison of numerous data, but one should consider that different tests were administered with different items in each study, different neuro-imaging devices were used and, most importantly, the group of participants was different in each study. That is, the participants included to test the timing and/or localization of one stage were different from the group tested for another stage. Another drawback is that the stage of phonological encoding is not included in Indefrey′s [3] model.

Den Hollander et al. [4] measured the time course in one group of younger (17–28 years old) and one group of older (40–65 years old) adults with tests specifically tapping into each of the processes mentioned in the previous section. For this approach, EEG was used. Interestingly, neither the time course, nor the scalp distributions differed between the groups. The time course of lemma and lexeme retrieval were highly comparable to those given by Indefrey [3]. Additionally, den Hollander et al. [4] provided data for phonological encoding, which takes place between 350–415 ms post stimulus presentation.

These data allow the authors of this paper to create Figure 3 in order to represent the localization as well as time course of naming a picture of an object or animal. Notice that the process we delineated here is just a very minor part of the entire speech and language production: there are no verbs, grammar or sentences represented yet. However, this model is a good start for personalizing medical treatment.

## 4. Navigated Transcranial Magnetic Stimulation (nTMS) for Language Testing

In a recent study, Ohlerth et al. [5] reported data on testing for verb and noun production in healthy adults, using nTMS. For this, the *Verb And Noun test for Peri-Operative testing (VAN-POP* [1]; see Figure 1 above) was administered to healthy adults while the cortex was stimulated with nTMS. New in this study is that (a) both verbs and nouns were tested in sentence context and (b) both hemispheres were stimulated. The results of this experiment are shown in Figure 4.

These positive language sites were subsequently used as seeding points for fiber tracking. The left ventral stream (Inferior Fronto-Occipital and Inferior Longitudinal Fascicle) is significantly better visualized, and, hence, more involved in action naming than in object naming, again, showing that action naming (in sentence context with an inflected verb, as assessed by the VAN-POP) is more taxing than object naming [6].

Notice that these are group results: at the individual level there may be areas and tracts that are only involved in action or object naming. This is even more likely in glioma patients. As said, considerable functional reorganization may occur in these patients: language functions may shift and this may be to different sites for verbs and nouns. The result, in laymen’s terms, might be that if you only test for nouns, you may well cut out verbs. Additionally, word retrieval may be delayed in these patients, meaning that the time course given in Figure 3 is not appropriate for this group. In the next section, we will present two glioma patients who underwent presurgical language testing with nTMS to illustrate the aforementioned individual (re)organization.

## 5. Two Case Studies

For a full description of the clinical workflow and details on the assessments, see Ohlerth et al. [7]. In addition to the VAN-POP action-naming items, a separate set of object naming items were used in this clinical workflow. Characteristics of the items, however, largely overlap with the VAN-POP and, therefore, do not alter the mapping approach.

### 5.1. Case 1

Case 1 (see case 6, Ohlerth et al. [7]) presented with a presumed low-grade glioma in the inferior temporal gyrus, most easily accessed during surgery via medial temporal regions. Language mapping in these regions was, therefore, particularly important in order to avoid damage to lemma and lexeme retrieval.

Preoperatively, the patient performed in a similar way to non-brain-damaged individuals during object naming; he showed a slight impairment during action naming compared to control data. The latter may have been caused by the tumor already preoperatively. During mapping with nTMS, positive areas for object naming were elicited mostly in the parietal lobe, while action naming seemed to rely on a larger network in all three lobes, particularly in the superior temporal gyrus. At the subcortical level, the networks for nouns and verbs differed as well.

The eloquence of the temporal lobe was once more tested intraoperatively with both tasks, and it turned out that the spots predicted with action naming during nTMS were now also positive areas under DES and, henceforth, avoided during resection.

On day 3 after the operation, the patient displayed a dissociated pattern of skills concerning the two tasks: his scores decreased significantly for object naming to only 28/80 correct items, while action naming was also impaired, but significantly less. Scans confirmed that the reason for this dissociation was a damage to the subcortical network of object naming, but less so for the network of action naming. While this impairment most likely improved over time, the patient was visibly relieved that while retrieval of nouns was troublesome, small sentences around the verb could be formed.

### 5.2. Case 2

Case 2 (see case 7, Ohlerth et al. [7]) presented with an insular lesion, a presumed low-grade glioma, extending into the temporal lobe, once more most easily accessed through temporal gyri. The patient showed no impairment preoperatively on either object or action naming. During object naming under nTMS language mapping, nearly exclusively articulatory errors could be elicited, mainly in the frontal lobe. During action naming under nTMS, however, many more errors, including retrieval errors, were found that spanned over parietal and the crucial temporal regions. Subcortically, in line with this, divergent white matter networks were visualized for the two tasks. Again, intraoperative mapping with DES confirmed the positive areas predicted by nTMS mapping with action naming on the superior temporal gyrus and led to a surgery preserving this area.

Postoperatively, the patient showed no difference in object naming compared to preoperatively. However, action naming scores dropped significantly, manifesting in many semantic paraphasias, when the patient failed to retrieve the correct verb and instead tried to circumscribe it. Scans confirmed that this reverse dissociation was likely due to slight damage of the subcortical white matter network of action naming, but not of object naming.

These cases illustrate the double dissociation of separate cortico–subcortical networks of object and action naming, where damage to one does not entail damage to the other skill.

## 6. Discussion

There are some noteworthy findings in this series of studies: (1) the involvement of the right hemisphere, in both action naming and object naming; (2) the relationship between language testing with nTMS and the time course of word production; (3) the difference between verbs and nouns; (4) the benefits of nTMS (versus DES). Finally, we will discuss the clinical implications.

### 6.1. Bilateral Representation of Naming

The first and rather surprising finding is that there is no difference in the number of overall errors between the left and right hemisphere, whereas the left hemisphere is supposed to be dominant for language. Errors were classified to tap into the different processes: conceptual–lexico–semantic errors (lemma and lexeme level), phonological encoding and articulation (grammatical errors were also counted, but are ignored here, because they hardly occurred). There was no difference in error pattern between the left and right hemisphere and the majority of errors were lemma and lexeme retrieval errors (30–40%) and phonological-articulatory errors (30–40%) in the left hemisphere; 40–50% lemma and lexeme retrieval errors (40–50%) and phonological–articulatory errors (30–40%) in the right hemisphere. Since the stimulus presentation was synchronized with the stimulation, it may very well be the case that concept retrieval was disrupted by the nTMS pulse, which is a non-language process preceding language production and, hence, no difference between errors elicited in the left and right hemisphere is expected at the level of conceptualization. Disruption of conceptualization may result in an inability to find the correct lemma (and thus, lexeme) or activation of an incorrect, probably meaning-related lemma (and, thus, lexeme). This effect is in line with previous fMRI research on non-verbal processing of objects and actions: in these tasks, activation of frontal and posterior parts of the left and right hemisphere has been observed [8,9,10]. The same holds for articulation errors: articulation is represented bilaterally and is a motor rather than language function; hence, no difference between the hemispheres, nor between word class is expected.

### 6.2. Actions/Verbs versus Objects/Nouns

Another important finding is that more lemma and lexeme retrieval errors are made on action naming than on object naming, meaning that action naming turns out to be more easily disrupted by nTMS. This means storage and activation of actions/verbs differs from that of objects/nouns. This is not very surprising. We know already from the aphasiological literature that verbs and nouns can be differentially affected after brain damage [11,12,13]. Additionally, several fMRI and MEG studies suggest that the areas involved in action naming only partially overlap with those involved in object naming [14,15]. Curiously, at the group level, we cannot pinpoint areas specifically recruited for either action naming or object naming. This is contrary to the view of embodied cognition, that suggests that action verbs are stored around the left (pre)motor area, whereas object nouns are stored or processed more posteriorly in the left hemisphere [16,17], and confirms the findings of Zubacaray et al. [18]. However, again for the group of healthy speakers, at the subcortical level, the left ventral stream is more involved in action naming than object naming. Together, these findings suggest that verb production requires a larger, but not entirely different, cortical and subcortical network. This is not surprising considering that actions involve more entities than nouns (e.g., *eating* requires an eater and something that is eating, whereas an *apple* is just an apple) and that, in sentence context, verbs require more grammatical processing (inflection for time, agreement and number).

### 6.3. nTMS and the Time Course of Word Production

As argued above, part of the lemma and lexeme retrieval errors may have occurred because the correct concept could not be retrieved. The fact that we used nTMS with a stimulus onset time of 0 ms (picture presentation and stimulus were synchronized), in line with the commonly used protocol used in preoperative mapping, may have caused this [19]. Hence, this may not be the best way to interrupt the naming process. Figure 3 above illustrates both localization and timing of the various consecutive processes. This suggests that better tuning of the nTMS stimulus, appropriate to the steps of the naming process may be required. For example, when the middle temporal gyrus is stimulated, where the lemmas of the words are supposed to be accessed, a stimulus delay of around 200 ms may be much more effective in disrupting the process of lemma retrieval than stimulating at 0 ms. In this way, more accurate maps of areas relevant in action and object naming can be acquired in non-brain-damaged speakers as well as in tumor patients. Unfortunately, we do not know the time course, nor the localization for the process of action naming.

### 6.4. Benefits of nTMS versus DES

DES is used directly on the cortical and subcortical tissue whereas nTMS is applied through the skull and can only inhibit cortical tissue (which can then, indirectly be used as seeding point for reconstruction of the subcortical tracks). This makes DES more reliable. There are several drawbacks of DES as well: firstly, not every brain tumor patient can tolerate the procedure, and secondly, only small areas, those exposed in the vicinity of the tumor during a craniotomy, are stimulated, meaning that DES mapping is hardly ever applied in both hemispheres. Apart from that, DES cannot be used experimentally due to its invasive nature, so it is hard to validate. nTMS, on the contrary, is tolerated very well and can be used in healthy speakers for experimental purposes on a large part of the cortical surface.

Until now, there have not been many studies that can validate the results on action and object naming, but we encourage researchers to replicate our study and to develop alternative tests, not only for language but for other cognitive functions as well. nTMS results have been compared to those of DES within patients, and showed that its specificity and sensitivity are high (90%) for delineation of the negative sites, but not yet ideal for the positive sites (40–70%) [20,21,22]. This means that DES should still remain the gold standard, but presurgical language mapping with nTMS has shown that presurgical mapping with nTMS allows for smaller craniotomy and has the added benefit of informing the neurosurgeon on how to approach the tumor [23].

The above-outlined problem with timing of stimulation is relevant for nTMS, but not for DES. DES usually lasts up to 4 s, meaning it disrupts the entire process from conceptualization to articulation. This means that no fine tuning is possible with DES, but since it is hardly ever used experimentally, it is not necessarily a drawback.

### 6.5. Clinical Implications

The case studies described above show the importance of presurgical testing for language areas. Verbs and nouns are represented differently, so when only one word class is assessed, the other one may be affected by brain surgery. More importantly, gliomas infiltrate healthy, functional tissue that may host language. Since they grow very slowly, functional reorganization may take place: (language) functions can shift to adjacent areas, but contralateral areas may compensate as well. Usually, the tissue taking over a function is less efficient at its new function than the original area was, meaning that the process of word production may be (slightly) slower than normal. nTMS allows to adapt stimulation timing to each individual patient. Additionally, one complete hemisphere (or even both hemispheres) can be scanned to obtain a complete picture of the representation of word retrieval.

We have shown that it is important to test more than nouns as single words. We started with verbs and nouns, both in sentence context, but since nTMS is tolerated so well, more language and cognitive functions, adapted to the wishes and interests of the patients, can be tested in several sessions. Moreover, in the present multilingual world, patients can be tested in each of the languages they speak. This is crucial since testing only one language in a bilingual patient during surgery may result in (partial) loss of the other.

These prospective applications underline that presurgical language testing with nTMS is still in its infancy. Yet, the current studies suggest that it is a promising tool for personalized medicine in brain tumor patients as it is safe, quick and reliable, and can investigate many more areas than can be accessed intraoperatively.

## Figures and Tables

**Figure 1 jpm-13-00376-f001:**
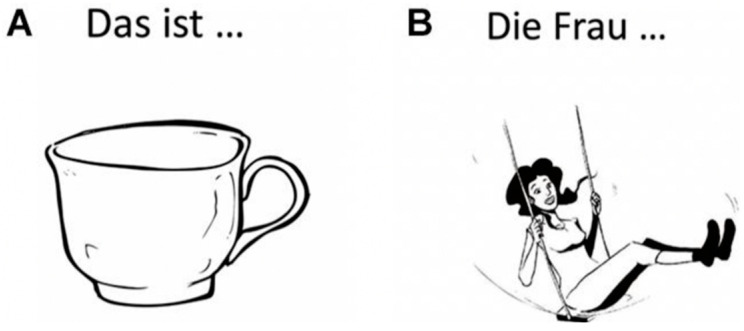
Example of a noun (**A**): Das ist … *eine Tasse:* ‘this is … a cup’) and a verb (**B**): Die Frau … *schaukelt:* ‘the woman … swings’) item of the VAN-POP^1^ (artwork: Victor Xandri Antolin; ©University of Groningen).

**Figure 2 jpm-13-00376-f002:**
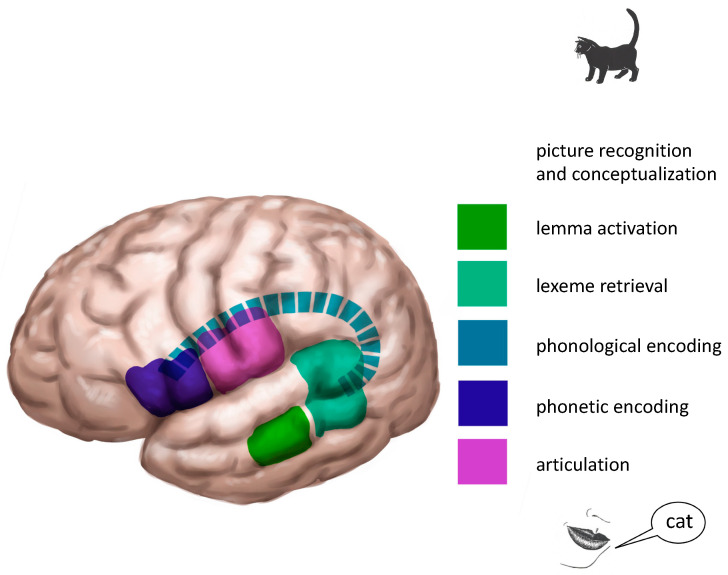
Localization of the naming process in the left hemisphere. Notice that for the preceding processes of ‘picture recognition and conceptualization’, large parts of both hemispheres are activated (not shown). Articulation is also a bilateral process. Based on Indefrey [3]. © Gert van Dijk.

**Figure 3 jpm-13-00376-f003:**
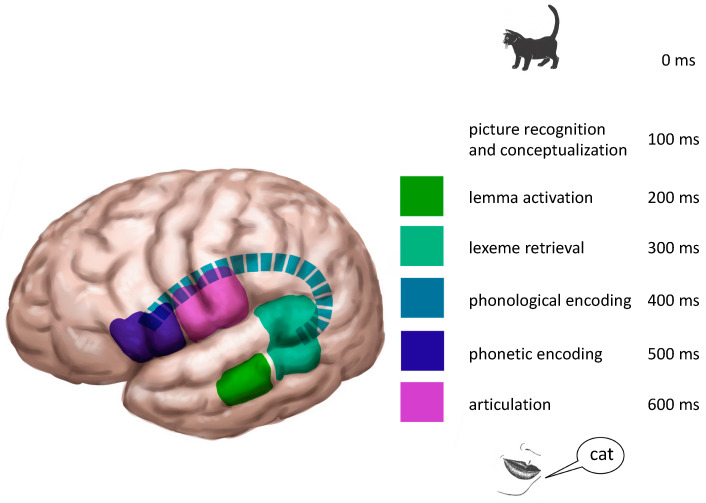
The timing process of object naming. Data based on Indefrey [3] and Den Hollander et al. [4]. ©Gert van Dijk.

**Figure 4 jpm-13-00376-f004:**
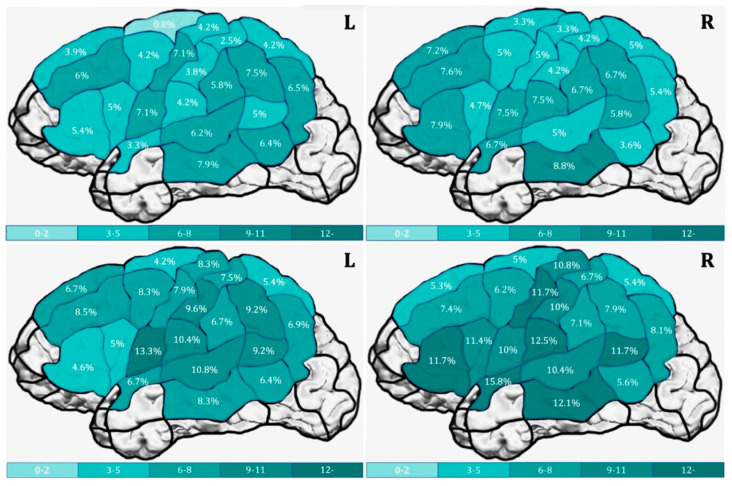
The percentage of errors (as color-coded) after nTMS stimulations, produced by a group of 20 non-brain-damaged young adults on object naming (**upper panel**) and action naming (**lower panel**) part of the VAN-POP [1] (figure from Ohlerth et al. [5]).

## Data Availability

The raw data supporting this article’s conclusions will be made available by the authors, without undue reservation.

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
