# Peer review of "Presurgical Language Mapping: What Are We Testing?"

_jpm, 2023, doi:10.3390/jpm13030376_

Round 1

Reviewer 1 Report

The authors present a very interesting report on their experience with an in-house developed language-mapping paradigm (VAN-POP), which, to my understanding, addresses object and action naming for TMS and DES mapping of speech/language.

Minor concerns:

1)      The title suggests a comprehensive overview of language mapping, whereas the manuscript deals exclusively with naming objects and actions. Could you please comment?

2)      The authors claim that TMS allows "cortical and subcortical areas to be identified" [with TMS]. Later, the authors correctly state that TMS inhibits the area "directly under the skull" [and not the subcortex]. Please clarify.

3)      The claim that neurosurgeons aim to "preserve as much functional tissue as possible" seems to suggest that there is also principally non-functional brain tissue. It might be more accurate to say that neurosurgeons aim to save tissue that is critical to a particular function (and injury would therefore not be compensable)?

Major concerns

4)      The authors report equal error rates in both hemispheres. This seems not to reflect decades of experience with awake intraoperative mapping and surgical outcomes. The authors suggest that this is because TMS with an on-set delay of 0ms interferes with bilateral processes like concept retrieval and articulation. This could be true for a subset of errors (mainly related to posterior brain areas), but remains speculative and contradicts not only DES but also other functional imaging studies. The alternative explanation is a lack of TMS efficacy leading to nonspecific, arbitrary observations highlighted by a low frequency of errors and a rather homogeneous pattern of errors. Please address these concerns. Figure 2 refers exclusively to the left hemisphere for a reason

5)      The claims about localization and timing of speech processing refer to a particular publication with specific claims about a sequential pattern of rhythmic steps of language processing (the authors acknowledged this). Nevertheless, there are other claims about timing and localization that should have their place in this methodological paper in my view.

6)      The authors reference their papers, where the example cases and the methodology have been recently published. What is the additional content compared to the authors previous work (e.g. Ref. 7)?

Author Response

Response to Reviewer 1

We would like to thank the reviewer for carefully reading our paper. We will go through the review point by point.

The authors present a very interesting report on their experience with an in-house developed language-mapping paradigm (VAN-POP), which, to my understanding, addresses object and action naming for TMS and DES mapping of speech/language.

The VAN-POP is not really 'in house' language test, it has been adapted to several other languages and is used in clinics worldwide. We don't feel to need to mention that.

The title suggests a comprehensive overview of language mapping, whereas the manuscript deals exclusively with naming objects and actions. Could you please comment?

We tested retrieval of inflected verbs and article+nouns in sentence context and the participants were supposed to produce an (admittedly short) sentence. This requires not only the retrieval of a verb or a nouns, but grammar as well. Also, the procedure described in the experimental sections is known as 'presurgical language testing'. That is why we like to keep the title as it was.

The authors claim that TMS allows "cortical and subcortical areas to be identified" [with TMS]. Later, the authors correctly state that TMS inhibits the area "directly under the skull" [and not the subcortex]. Please clarify.

It is true that TMS only inhibits tissue directly under the skull and in the abstract we have omitted the 'cortical areas'. On page 10, we explain that we used the positive langauge sites as seeding points for reconstruction of the subcortical tracts. This is a recognized and reliable method for visualizing these subcortical tracts. So, although not directly stimulated, the tracts are reconstructed in order to analyze their involvement in the tasks we administered.

The claim that neurosurgeons aim to "preserve as much functional tissue as possible" seems to suggest that there is also principally non-functional brain tissue. It might be more accurate to say that neurosurgeons aim to save tissue that is critical to a particular function (and injury would therefore not be compensable)?

The reviewer is right, of course, in the healthy brain all tissue is functional. Here we used 'functional tissue' as opposed to tumor tissue. 'Functional' has been changed to 'healthy' in the present version.

is that this sissue should be spared and the tumor tissue should be resected. We rephrased this as follows: "When low-grade, they grow very slowly, allowing motor and cognitive functions to migrate to other brain areas. During tumor resection, the neurosurgeon aims to save as much of this functional tissue as possible, but cannot rely on classical localization maps, because of this functional plasticity. "

The authors report equal error rates in both hemispheres. This seems not to reflect decades of experience with awake intraoperative mapping and surgical outcomes. The authors suggest that this is because TMS with an on-set delay of 0ms interferes with bilateral processes like concept retrieval and articulation. This could be true for a subset of errors (mainly related to posterior brain areas), but remains speculative and contradicts not only DES but also other functional imaging studies. The alternative explanation is a lack of TMS efficacy leading to nonspecific, arbitrary observations highlighted by a low frequency of errors and a rather homogeneous pattern of errors. Please address these concerns. Figure 2 refers exclusively to the left hemisphere for a reason

We are afraid we do not entirely agree with the reviewer on this point. Articulation is represented bilaterally in the frontal areas of the brain, we don't think the reviewer disagrees on this. Many fMRI studies show bilateral representation of conceptual knowledge of verbs, both frontal and posterior (Hayes et al., Assmus et al., Schubotz et al; all cited in the new version). It is true that bilateral involvement in language tasks has not been reported in DES studies, but then, DES is hardly ever applied to both hemispheres. So, we do not agree that there is a 'lack of TMS efficacy'.

Figure 2 only involved the left hemisphere, because this hemisphere is shown by Indefrey, on whose excellent paper our description is largely based. We explicitly mention in the text (with motivation) and in the figure legend that conceptualization and articulation are represented in both hemispheres. In the Discussion that we added on request of the other reviewer, we address the differences between DES and nTMS (p14-15).

The claims about localization and timing of speech processing refer to a particular publication with specific claims about a sequential pattern of rhythmic steps of language processing (the authors acknowledged this). Nevertheless, there are other claims about timing and localization that should have their place in this methodological paper in my view.

This 'particular publication' concerns a meta-analysis of about all neuro-imaging papers done on noun production. We are not aware of other claims about timing and localization, other than those that differ minimally from the generally accepted pattern. We apologize for not understanding what the reviewer means by 'a sequential pattern of rhythmic steps of language processing'.

The authors reference their papers, where the example cases and the methodology have been recently published. What is the additional content compared to the authors previous work (e.g. Ref. 7)?

We apologize for ticking the box of a 'research paper'. Our aims was to write a tutorial, but that option was not given. We will now tick the box of 'commentary' Our aim was to give an overview of our work on presurgical language testing in which we combine the model sketched by Indefrey and our reseach papers on presurgical language mapping, showing that nTMS, although still in its infancy, is a promising tool. We suggest that researchers should make more use of its possibilities to fine-tune stimulation and language tasks (or other cognitive tasks) in order to get better results with respect to sensitivity and specificity. This is now mentioned in the new Discussion.

Reviewer 2 Report

In the present article, the authors provide a good overview of the relevance of preoperative speech mapping using nTMS and of the anatomical-functional process of recognizing and naming objects and actions. The article is well written, clearly structured, of clinical relevance and interesting to read.

However, one point is not clear to me: the publication is marked as an original article but rather corresponds to a summary or review of the authors own previously published results and work. Here, the authors should please clearly work out which new scientific findings result from the present publication that have not already been reported in the referenced publications (1,4,5,6,7). Furthermore, in my opinion, a detailed discussion (including references) of the described results is missing. This would give the reader an even better insight into the topic and the relatively new knowledge in the context of the existing literature. Furthermore, the possibilities and the importance of this for personalized medicine should be emphasized more strongly. The conclusion should be shortened and presented more concisely.

Author Response

We thank the reviewer for the encouring words and the very helpful comments.

We apologize for ticking the wrong box. Our intention was not to submit an original research paper, but rather a tutorial. However, that possibility was not offered. the editor suggests to mark it as a Commentary, which is what we now did. It is true, it is a review of several of our research studies, embedded in a theory about word production.

We now have added a Discussion, in which we related our work to other research papers. In the Discussion we address 1) the involvement of the right hemisphere, in both action naming and the object naming; (2) the difference between nouns and verbs; (3) the relation between language testing with nTMS and the time course of word production; (4) the benefits of nTMS (versus DES); (5) clinical implications (in which we specifically address the merits of our paper to personalized medicine).

Round 2

Reviewer 1 Report

Thank you for adressing all my concerns.

Reviewer 2 Report

Knowing the authors' explanation ("Our intention was not to submit an original research paper, but rather a tutorial. However, that possibility was not offered. the editor suggests to mark it as a Commentary, which is what we now did. It is true, it is a review of several of our research studies, embedded in a theory about word production") and after comprehensive revision of the discussion, I recommend acceptance of the paper as a tutorial.